# New Insights into *MdSPS4*-Mediated Sucrose Accumulation under Different Nitrogen Levels Revealed by Physiological and Transcriptomic Analysis

**DOI:** 10.3390/ijms232416073

**Published:** 2022-12-16

**Authors:** Xuejing Cao, Wenfang Li, Ping Wang, Zonghuan Ma, Juan Mao, Baihong Chen

**Affiliations:** College of Horticulture, Gansu Agricultural University, Lanzhou 730070, China

**Keywords:** apple, nitrogen, physiology, transcriptome, sugar metabolism, *MdSPS4*

## Abstract

Nitrogen nutrition participates in many physiological processes and understanding the physiological and molecular mechanisms of apple responses to nitrogen is very significant for improving apple quality. This study excavated crucial genes that regulates sugar metabolism in response to nitrogen in apples through physiology and transcriptome analysis, so as to lay a theoretical foundation for improving fruit quality. In this paper, the content of sugar and organic acid in apple fruit at different developmental periods under different nitrogen levels (0, 150, 300, and 600 kg·hm^−2^) were determined. Then, the transcriptomic analysis was performed in 120 days after bloom (DAB) and 150 DAB. The results showed that the fructose and glucose content were the highest at 120 DAB under 600 kg·hm^−2^ nitrogen level. Meanwhile, different nitrogen treatments decreased malate content in 30 and 60 DAB. RNA-seq analysis revealed a total of 4537 UniGenes were identified as differentially expressed genes (DEGs) under nitrogen treatments. Among these DEGs, 2362 (52.06%) were up-regulated and 2175 (47.94%) were down-regulated. The gene co-expression clusters revealed that most DEGs were significantly annotated in the photosynthesis, glycolysis/gluconeogenesis, pyruvate metabolism, carbon metabolism, carbon fixation in photosynthetic organisms and plant hormone signal transduction pathways. The key transcription factor genes (*ERF*, *NAC*, *WRKY*, and *C2H2* genes) were differentially expressed in apple fruit. Sugar and acid metabolism-related genes (e.g., *HXK1*, *SPS4*, *SS2*, *PPC16-2,* and *MDH2* genes) exhibited significantly up-regulated expression at 120 DAB, whereas they were down-regulated at 150 DAB. Furthermore, the *MdSPS4* gene overexpression positively promoted sucrose accumulation in apple callus and fruit. In conclusion, the combinational analysis of transcriptome and the functional validation of the *MdSPS4* gene provides new insights into apple responses to different nitrogen levels.

## 1. Introduction

Nitrogen is one of the three major elements of fruit tree nutrition [1]. The appropriate application of nitrogen fertilizer can not only increase the photosynthetic rate and photosynthetic area of leaves, but also promote flower bud differentiation, improve fruit quality, and increase average fruit weight. Excess or deficiency of nitrogen fertilizer affects the growth and development of fruit trees [2]. Nitrogen metabolism is closely related to carbon metabolism, carbon and nitrogen balance, which are necessary for optimal plant growth and development [3]. Numerous genes are regulated by N and C/N interactions, which was confirmed in *Arabidopsis* [4]. Meanwhile, the signaling network of nitrate and sugars interacts strongly [5]. N nutrients can be provided in inorganic (nitrate [NO_3_^–^] and ammonium [NH_4_^+^]) or organic (amino acids and urea) forms. However, nitrate is the dominant source of N both in agriculture and many natural systems [6,7]. 

Sugar metabolism provides a carbon source and energy for plants. The accumulation of sugar in plants can regulate the source-sink relationship and the utilization efficiency of sugar in cells, which can improve plant photosynthesis and biomass yield [8]. In addition, the composition and accumulation of sugar in cells are also important factors affecting fruit quality [9]. Higher plants usually use solar energy to convert carbon dioxide in photosynthetic leaves into organic carbon. For most plants, sucrose is the end product of photosynthesis for translocation through the sieve element/companion cell complex of the phloem from the source to heterotrophic sinks. This sucrose-controlled phloem transport is the dominant pathway for water, nutrients, and signaling molecules, which are imported to meristematic sinks, including the shoot and root apical meristems [10]. At the cellular level, sucrose is the crucial carbon source for growth, development, and response to the adversity of plants. Sucrose metabolism produces hexose (Hex), which is required for energy production and synthesis of starch, fructose, cellulose, protein, and antioxidant compounds [11].

In general, the sucrose metabolism in plants is involved in several enzymes such as sucrose synthase (SuSy; EC2.4.1.13), sucrose phosphate synthase (SPS; EC2.4.2.14), and invertase (INV; EC3.2.1.26). SPS is the pivotal enzyme for the synthesis of sucrose from uridine diphosphate-glucose (UDPG) and fructose-6 phosphate (F6P) to sucrose-6-phosphoric acid (S6P), which is irreversibly converted to sucrose with sucrose phosphatase (SPP) [12]. Genes-encoding SPS was first discovered in wheat [13]. Subsequently, *SPS* has been cloned from various plants, including rice, maize, sorghum [14], *Arabidopsis* [15], tobacco [16], tomato [17], sugarcane [18], and poplar [19]. These previous studies indicated that the activity of SPS played a positive regulatory role in sucrose accumulation, and that the total sugar is mainly in the form of sucrose in sorghum stalks [13,20]. *SPS* overexpression in tobacco affected carbon allocation and the carbohydrate metabolism, accelerated plant growth, and increased sucrose synthesis in the old leaves of transgenic plants [21]. Moreover, *SPS* overexpression has increased the sucrose/starch ratio and photosynthetic rate in the leaves of transgenic *Arabidopsis* [15] and tomato [17]. Meanwhile, *SPS* overexpression also increased sucrose unloading in tomato fruit [17]. The effects of *SPS* overexpression on plant growth and biomass were also examined in transgenic *Arabidopsis* [22] and poplar [19], tobacco [16], and brachypodium [18].

Apple is rich in minerals and vitamins and is planted in temperate regions of the world. Fruit quality is an indispensable indicator in apple production. The carbohydrates in apple are vital factors affecting the fruit quality, taste, and the formation of other secondary metabolites [23]. In the loess plateau region, the main cultivation pattern is vigorous rootstocks grafting short branch varieties [24]. Meanwhile, the planting area of the ‘Oregon Spur Delicious’ apple in the Tianshui City of Gansu Province is more than 66,666.67 hectares, which is the largest marshal apple cultivation area in the world. ‘Oregon Spur Delicious’ apple is the fifth generation of marshal apple with short shoots, which has the characteristics of bright skin color, crisp taste, tender fleshyes, and strong aroma [25]. The loess plateau region has low organic matter content, poor soils, and is prone to nutrient loss [26]. Under these ecological conditions, higher N fertilizer application can significantly increase the yield and quality of apples. Therefore, it is very significant to elucidate the mechanism of different nitrogen levels on the apple sugar metabolism. 

Previous studies have focused on the effects of nitrogen fertilization on apple physiology [27,28,29], whereas the reports on the molecular mechanisms of nitrogen regulating sugar metabolism in apple are not characterized. Here, RNA-Seq transcriptome analysis has been conducted to characterize the dynamic responses of different nitrogen levels toward apple fruit. The results of the RNA-Seq analysis have identified several signaling pathways and candidate genes related to sugar and organic acid. In addition, overexpression of *MdSPS4* increased sucrose accumulation in apple fruit and callus. This provides a novel reference for molecular breeding efforts linked to improving fruit quality.

## 2. Results and Discussion

### 2.1. Effects of Different Nitrogen Levels on Soluble Sugar

The most prominent carbohydrates in apple fruit are fructose and glucose, followed by sucrose and sorbitol. Here, with the increase of days after bloom (DAB), the fructose content in apple fruit generally increased first and then decreased (Figure 1A). At 30 DAB, the fructose content of the T1–T3 treatments was significantly higher than that of the T0 treatment. Moreover, the fructose content was highest at 120 DAB compared to other periods, and the fructose content of the T2 and T3 treatments was significantly higher than the T0 treatment. The glucose content of the T1 treatment was generally higher than that of the T0 treatment (Figure 1B). At 120 DAB, the glucose content of the T2 and T3 treatments was significantly higher than the T0 treatment. At 150 DAB, the glucose content of the T1–T3 treatments was significantly higher than the T0 treatment. The sucrose content in apple fruit generally increased as the DAB increased, and the sucrose content of the T0–T3 treatments was too low to be detected at 30 DAB (Figure 1C). The sucrose content of the T1–T3 treatments was higher than that of the T0 treatment, except for 150 DAB. The sucrose content was highest at 150 DAB compared to other periods. With the increase of DAB, the sorbitol content in apple fruit generally showed a downward trend, and the sorbitol content in the T2 treatment was higher than that in other treatments (Figure 1D). The sorbitol content was the highest at 30 DAB compared to other periods, and the sorbitol content of the T1 treatment was significantly higher than the T0 treatment.

Different nitrogen levels have been shown to greatly influence the sugar composition [30]. This research has shown that the sucrose, glucose, and fructose content in citrus fruit would be higher with the higher N until the N3 treatment (1.81 kg·y^−1^) slightly decreased [25]. Wang et al. [31] indicated medium to high nitrogen (600 kg·hm^−2^) and high nitrogen (800 kg·hm^−2^) decreased sugar content of the apple fruit by 16.05 %. Meanwhile, Sokri et al. [32] revealed that the quality of the apple fruit deteriorated under high N. This study showed that the sugar content in the ‘Oregon Spur 2′ apple varied with different development periods and different nitrogen levels. Fructose and glucose content was highest at the fruit expansion period under 600 kg·hm^−2^ (Figure 1A,B), sucrose content was highest at the fruit ripening period under 150 kg·hm^−2^ (Figure 1C), and sorbitol content was the highest at the young fruit period under 150 kg·hm^−2^ (Figure 1D). The results indicated that different nitrogen levels increased the fructose, glucose, sucrose, and sorbitol content of apple compared to the control at different development periods. The results obtained in this paper are similar to those in citrus [25], tomatoes [17], melon [33], and date palm [34].

### 2.2. The Effects of under Different Nitrogen Levels on Organic Acids

With the increase of DAB, the malic acid content first increased and then decreased (Figure 2A). The malic acid content of the T0 treatment was highest at 60 DAB, the malic acid content of the T1 and T2 treatments was highest at 120 DAB, and the malic acid content of the T3 treatment was highest at 90 DAB. Moreover, the malic acid content of the T1–T3 treatments at 30 DAB was significantly higher than the T0 treatment. The quinic acid content decreased continuously with the increase in DAB, and the young fruit stage was the highest and the mature stage was the lowest (Figure 2B). At 30 DAB, the quinic acid content of the T1–T3 treatments was significantly higher than the T0 treatment. Meanwhile, the quinic acid content of the T3 treatment at 150 DAB was significantly higher than the T0 treatment. The ascorbic acid content decreased continuously as DAB increased (Figure 2C). The citric acid content changed greatly in different development periods (Figure 2D). At 30 DAB, the citric acid content of the T1 treatment was highest. At 60 and 120 DAB, the citric acid content of the T0 treatment was at its highest, and the citric acid content of the T0 treatment was significantly higher than that of the T1–T3 treatments. 

Malate is a pivotal metabolite, which is regulated by various signaling pathways. Physiological studies have shown that malate and nitrate have complex regulation mechanisms during plant growth and development [35]. While the molecular mechanism between nitrate and malate has remained unclear, some reports have demonstrated that nitrogen positively regulates fruit acidity [36], while others suggested nitrate level and fruit acidity have a negative relationship [37]. In this study, the malic acid content of different nitrogen levels was lower than the control at 30 and 60 DAB. Contrary to that, the malic acid content was higher than the control at 120 DAB (Figure 2A). In addition, the content of quinic acid, ascorbic acid, and citric acid in apple fruit under different nitrogen levels was lower than the control at 30 or 60 DAB (Figure 2A–C). This paper speculates that nitrogen fertilization had a greater effect on the organic acid content in apple fruit and significantly reduced the organic acid content during the young fruit period.

### 2.3. Transcriptome Sequencing Analysis

Eighteen RNA samples were harvested from non-nitrogen fruit (0 kg·hm^−2^) and nitrogen fruit (300 kg·hm^−2^ and 600 kg·hm^−2^) at 120 DAB (named S1-0, S1-2, and S1-3) and 150 DAB (named S2-0, S2-2, and S2-3), respectively. Then, cDNA libraries were constructed for RNA-seq analysis. The Q30 percentage was over 93.23%, the percentage of GC content was between 47.67% and 48.04% (Table 1). Mapped reads ranged from 88.36% to 91.62%, and the unique match ranged from 85.98% to 88.39%, indicating that the fruit samples and transcriptome data were considered reliable for further analysis.

### 2.4. DEGs Analysis at Different Development Stages under Different Nitrogen Levels 

After data compiling, 1481, 2650, 673, and 1348 genes were differently expressed at S1-2 vs. S1-0, S1-3 vs. S1-0, S2-2 vs. S2-0, and S2-3 vs. S2-0, respectively (Figure 3A, Appendix A). There were 821 up-regulated genes and 660 down-regulated genes in S1-2 vs. S1-0, 1364 up-regulated genes and 1286 down-regulated genes in S1-3 vs. S1-0, 231 up-regulated genes and 442 down-regulated genes in S2-2 vs. S2-0, 681 up-regulated genes and 667 down-regulated genes in S2-3 vs. S2-0 (Figure 3A). Of these DEGs, 1455, 2617, 650 and 1314 genes have been identified in SwissProt, GO, KOG, or KEGG databases, respectively (Figure 3B). Among the identified DEGs, 1368, 2468, 595, and 1199 genes had characterized functions, respectively (Appendix A). There were 265 DEGs that co-existed in two stages from the S1 to the S2 (Figure 3C). Based on biological functions and physiological characteristics, the 265 DEGs were divided into 10 categories: primary metabolism and energy (35), hormone biosynthesis (1), bio-signaling (36), cell morphogenesis (37), transcription factor (36), polynucleotide biosynthesis (5), translation (75), transport (16), secondary metabolism (13) and stress response (11) (Figure 3D). 

### 2.5. GO Classification Analysis 

DEGs annotated in the GO database were divided into three major functional categories: biological processes (BP), cellular components (CC), and molecular functions (MF) (Appendix A). BP category mainly includes cellular component organization or biogenesis (GO:0071840), single-organism process (GO:0044699), cellular process (GO:0009987), and metabolic process (GO:0008152). Binding (GO:0005488) and catalytic activity (GO:0003824) belong to the MF category. The CC category mainly includes cell part (GO:0044464), organelle (GO:0043226), membrane (GO:0016020), and cell (GO:0005623). 

### 2.6. KEGG Annotation Analysis

All 8679 unigenes were annotated in the KEGG pathway, which analyzed whether different nitrogen levels-responsive genes in apple were involved in some special pathways. A total of 372, 620, 100, and 247 DEGs were mapped to 100, 119, 62, and 96 KEGG pathways at S1-2 vs. S1-0, S1-3 vs. S1-0, S2-2 vs. S2-0, and S2-3 vs. S2-0, respectively. The top 20 KEGG pathways enriched in S1-2 vs. S1-0 and S1-3 vs. S1-0 were lighted in photosynthesis (ko00196), protein processing in endoplasmic reticulum (ko04141), carbon fixation in photosynthetic organisms (ko00710), plant hormone signal transduction (ko04075), carbon metabolism (ko01200) and glycolysis/gluconeogenesis (ko00010) (Appendix A). KEGG pathways enriched in S2-2 vs. S2-0 and S2-3 vs. S2-0 were mainly concentrated in plant-pathogen interaction (ko04626), carotenoid biosynthesis (ko00906), galactose metabolism (ko00052), protein processing in endoplasmic reticulum (ko04141), amino sugar and nucleotide sugar metabolism (ko00520), fructose and mannose metabolism (ko00051) (Appendix A). Overall, these results indicated that apple fruit may respond to different nitrogen levels by regulating various metabolic pathways. 

### 2.7. Expression Pattern and Functional Analysis of the DEGs in Different Nitrogen Levels 

According to the expression profiles, 3601 DEGs were classified into eight clusters by co-expression clustering (Figure 4, Appendix A). A total of 556 genes were classified into cluster 1, the expression of these DEGs was rapidly down-regulated from S1-0 to S1-3, indicating that their expression was repressed under the higher nitrogen levels, and mainly participated in “photosynthesis”, “fatty acid degradation” and “carbon fixation in photosynthetic organisms” pathways (Figure 4A,B). The expression of 351 DEGs in cluster 3 was higher in the S1-2 and the S1-3 than that in the S1-0, which were mainly enriched in “pyruvate metabolism”, “plant hormone signal transduction”, “fatty acid metabolism”, and “carbon metabolism” pathways. The expression of genes linked to cluster 5 was persistently down-regulated from S1-0 to S1-3, and the transcriptions were suppressed as the nitrogen levels increased, which was mainly enriched in the “photosynthesis”, “pentose phosphate pathway”, “glycolysis/gluconeogenesis”, “carbon metabolism” and “carbon fixation in photosynthetic organisms” pathways. Cluster 6 contained the most numerous genes, with a total of 792 genes, and the changes in expression levels were relatively stable from S2-0 to S2-3, mainly enriched in “RNA transport”, “pyruvate metabolism”, “protein processing in endoplasmic reticulum” and “fatty acid biosynthesis” pathways (Figure 4A,B). This study hypothesizes that different genes play different roles in apple fruit during the expansion and ripening periods.

### 2.8. Differentially Expression Transcription Factors (TFs) under Different Nitrogen Levels 

Plant transcription factors (TFs) are key hubs regulating plant growth and development. In this paper, 299 TFs differentially expressed in apple fruit under different nitrogen levels (Appendix A), and the top 10 TFs in S1-2 vs. S1-0, S1-3 vs. S1-0, S2-2 vs. S2-0, and S2-3 vs. S2-0 were selected. AP2/ERF is the most enriched TF in each group, followed by NAC, C2H2, bZIP, MYB, etc. (Figure 5A–D). Fourteen TFs, including ERF1 (MD04G1058000), ERF1A (MD06G1051800), ERF2 (MD01G1177000), ERF5 (MD04G1058200), bZIP43 (MD09G1078300), bZIP50 (MD06G1002100), MYB (MD09G1169000), MYB-related-1 (MD01G1090900), MYB-related-2 (MD09G1279300), C2H2 (MD15G1071400), Trihelix (MD14G1094300) and LOB (MD17G1078600), were co-enriched in four groups (Figure 5E). In S1-2 vs. S1-0 and S1-3 vs. S1-0, ERF1, ERF1A, ERF2, ERF5, bZIP50, MYB-related-2, Trihelix, C2H2, and LOB were up-regulated. Moreover, MYB-related-1, MYB-related-2, bZIP43, Trihelix, MADS-MIKC, and LOB were up-regulated in S2-2 vs. S2-0 and S2-3 vs. S2-0 (Figure 5F). Among these TFs, bZIP50 responds to endogenous stimulus and oxygen-containing compounds, MYB is involved in posttranslational modification, Trihelix responds to stress, regulation of the cellular metabolic process, single-organism developmental process, and meristem development. The results suggest these TFs could play an important role by regulating the expressions of target genes.

One study has reported that MaRAP2-4, an ERF TF from M. arvensis activates *AtSWEET10* in *Arabidopsis* by DRE and GCC boxes, which further influence carbohydrate allocation and accumulation [38]. In the present study, four ERF (ERF, ERF1A, ERF2, ERF5) TFs were identified at 120 and 150 DAB. Among them, the expressions of ERF2 and ERF5 was higher than other genes under nitrogen fertilization treatment. This study speculates that ERF2 and ERF5 might regulate some sugar metabolism/transport responsive genes through DRE or GCC motifs under nitrogen induction.

In maize, ZmNAC34 is a transcriptional repressor, which negatively regulates the expression of starch biosynthesis-related genes, thereby reducing the soluble solid content and starch content [39]. In watermelon, ClNAC68 is highly expressed in sweet flesh, which regulates the sugar content by repressing invertase activity in watermelon. Moreover, the knockout of ClNAC68 can reduce sugar accumulation, including sucrose, fructose, and glucose, and glucose was significantly lower in the ClNAC68 mutant fruit, especially at 26 days after pollination [40]. In this study, more target NAC TFs were identified at 150 DAB, and we speculate that nitrogen fertilizer has a great effect on NAC TFs during the fruit ripening period, which directly or indirectly regulate genes related to sugar metabolism.

Past studies have found that the apple MYB TF MdMYB1 directly activates expressions of the genes encoding tDT, V-ATPase subunits, and V-PPase, which modulates the malate accumulation in apple [41]. Another MYB TF, MdMYB73 could promote malate accumulation, which directly binds to the promoters of MdALMT9, MdVHA-A, and MdVHP1 [42]. In this paper, three MYB (MYB, MYB-related-1, and MYB-related-2) TFs were identified at 120 and 150 DAB, and the expression of MYB-related-1 was higher at 120 DAB than 150 DAB, especially under high N treatment. This study speculates that the higher nitrogen fertilizer can positively regulate the expression of MYB-related-1 during the fruit expansion period, and MYB-related-1 plays an important role in the accumulation of malic acid during this period.

### 2.9. DEGs Involved in Sugar Metabolism

A total of twenty-two DEGs were related to sugar metabolism (Figure 6). There were eleven genes involved in glycolysis/gluconeogenesis, including one glyceraldehyde 3-phosphate dehydrogenase (*GAPDH*), one Enolase (*ENO1*), two Fructose-bisphosphate aldolase (*FBA2-1*, *FBA2-2*), four Fructose-1,6-bisphosphatase (*FBP-1*, *FBP-2*, *FBP-3*, *FBP-4*) and three Hexokinase (*HXK-1*, *HXK-2*, *HXK4*). Genes associated with starch and sucrose metabolism including six sucrose-phosphate synthases (*SPS1-1*, *SPS1-2*, *SPS1-3*, *SPS2*, *SPS3*, *SPS4*), two fructokinases (*FRK-1*, *FRK-7*), one starch synthase (*SS2*) and two glucose-1-phosphate adenylyltransferases (*AGPS1*, *AGPS2*). Among them, *ENO1*, *FRK-7*, and *AGPS1* were down-regulated in S1-2 vs. S1-0, S1-3 vs. S1-0, S2-2 vs. S2-0, and S2-3 vs. S2-0. *HXK1-1*, *HXK1-2*, *HXK4*, *SPS1-1*, *SPS1-3*, *SPS3*, *SPS4-1*, *SPS4-2*, *SS2* and *AGPS2* were up-regulated in S1-2 vs. S1-0 and S1-3 vs. S1-0, whereas they were down-regulated in S2-2 vs. S2-0 and S2-3 vs. S2-0. In addition, *FBA2-2*, *FBP-1*, *SPS1-2*, *FRK-1* were up-regulated in S2-2 vs. S2-0 and S2-3 vs. S2-0. The results indicate that the sugar metabolism and transport are closely related to the expression levels of these DEGs. 

Sugar is relevant to the sweetness of fruit, which is the most attractive characteristic for consumers [43]. Furthermore, studies have reported that sugar may serve as important signals that modulate varieties of processes in plant physiology, including response to adversity, stress and fruit maturation [44]. The hexose sugar (glucose and fructose) products enter into the cells by means of monosaccharide transporters, then is phosphorylated by HXK and FRK and used for respiration or sucrose resynthesis. In this paper, the expressions of three *HXK* (*HXK1-1*, *HXK1-2*, *HXK4*) genes were found to be up-regulated at 120 DAB. In contrast, the expressions of three *HXK* genes were down-regulated at 150 DAB. This study hypothesizes that three *HXK* genes contribute to sucrose synthesis or accelerate plant respiration at the fruit expansion period, while at the fruit ripening period they promote hexose accumulation. In addition, the expression of *FRK-7* was down-regulated at 120 and 150 DAB. The expression of *FRK-1* was down-regulated at 120 DAB, while it was up-regulated at 150 DAB. This result indicates that *FRK-1* responded differently to nitrogen fertilization in different development periods and played different roles in fructose accumulation.

Sucrose can be resynthesized in the cytosol depending on sucrose synthase (SUS) from fructose and UDP-GLC. Sucrose can also be hydrolyzed to fructose and glucose for energy production relying on neutral invertase (NIN), transferred to the vacuole for storage, or even hydrolyzed by vacuolar acid invertase (AIN) [45]. *SPS* is viewed as the critical gene for sucrose accumulation in fruit ripening. This enzyme catalyzes the reversible transfer of a hexosyl group from UDP-glucose to D-fructose 6-phosphate to form UDP and D-sucrose-6-phosphate [46]. In this study, except for the *SPS1-2*, the expressions of other *SPS* genes were up-regulated at 120 DAB under different nitrogen levels, indicating that nitrogen fertilization positively regulated the expressions of these genes. Interestingly, the expressions of numerous genes were down-regulated at 150 DAB, except for the *SPS1-2*, indicating that nitrogen fertilization negatively regulated these genes during this period and that these genes responded differently to nitrogen fertilization between fruit expansion and ripening periods. This study hypothesizes that *SPS1-1*, *SPS1-3*, *SPS2, SPS3*, and *SPS4* genes could promote sucrose synthesis during the fruit expansion period and inhibit sucrose accumulation during the fruit ripening period.

### 2.10. DEGs Associated with Pyruvate Metabolism

Eight DEGs were excavated in the pyruvate metabolic pathway, including two pyruvate kinase (*PK2-1*, *PK2-2*), one dihydrolipoyl dehydrogenase (*DLD2*), two dihydro-lipoyllysine-residue acetyltransferases (*DLAT5-1*, *DLAT5-2*), two phosphoenolpyruvate carboxylase (*PPC16-1*, *PPC16-2*) and one malate dehydrogenase (*MDH2*) (Figure 7). All these eight DEGs were up-regulated in S1-2 vs. S1-0 and S1-3 vs. S1-0, while down-regulated in S2-2 vs. S2-0 and S2-3 vs. S2-0. The results suggest that these genes related to malate metabolism may play opposite roles during the fruit expansion and ripening stages.

Fruit acidity is an important index of fruit organoleptic quality, which with soluble sugar affects the flavor of the fruit, and is usually determined with titratable acid. The malic, tartaric and citric acids are the organic acids found in most ripe fruit and mostly accumulate in the vacuole [8]. Understanding the regulation mechanism of organic acid accumulation in fruit cells is an important aspect in improving fruit quality. Organic acids accumulate including metabolism and transport in the vacuoles of fruit cells. Organic acids are produced through two pathways: the tricarboxylic acid cycle in the mitochondria or conversion from oxaloacetate (OAA) in the cytosol [35]. Malate dehydrogenase (MDH) is a vital enzyme regulating malate metabolism, which catalyzes the reversible reaction between OAA and malate. It is reported that MDH in cytoplasm and mitochondria could regulate the accumulation of malate in the apple (*Malus domestica*) [47]. In this paper, the expression of the *MDH2* gene was up-regulated at 120 DAB. This study speculates that nitrogen fertilization positively regulated the expression of the *MDH2* gene during the fruit expansion period, thereby promoting the rapid accumulation of malic acid. This result is consistent with different nitrogen levels having the higher malic acid content compared with the control during this period [47]. In contrast, the expression of the *MDH2* gene was down-regulated at 150 DAB, and this study would assume that nitrogen fertilization negatively regulated the expression of the *MDH2* gene during the fruit ripening period. Interestingly, during this period, the content of malic acid in the nitrogen fertilization treatment was also lower than that in the control, suggesting that nitrogen fertilizer had different effects on the accumulation of malic acid by regulating the expression of genes related to acid metabolism in different stages of apple development.

### 2.11. RNA-Seq Expression Validation by qRT-PCR

qRT-PCR analysis was used to ensure the reliability of transcriptome data. Twenty DEGs closely associated with sugar and acid metabolism were selected as targets (Figure 8), including *GAPDH* (MD06G1148800), *ENO1* (MD06G1208300), *FBA2-1* (MD00G1040000), *FBA2-2* (MD10G1063600), *HXK1* (MD09G1202200), *HXK4* (MD03G1170700), *SPS2* (MD04G1013500), *SPS3* (MD10G1002400), *SPS4* (MD10G1002300), *SS2* (MD00G1130900), *AGPS1* (MD08G1027900), *AGPS2* (MD17G1122900), *PK2-1* (MD01G1212800), *PK2-2* (MD13G1202500), *DLD2* (MD16G1157500), *DLAT5-1* (MD05G1040900), *DLAT5-2* (MD10G1046900), *PPC16-1* (MD03G1242000), *PPC16-2* (MD11G1261900), *MDH2* (MD06G1193800). The result confirmed that qRT-PCR expression patterns were in preferable agreement with the RNA-Seq trend. 

### 2.12. MdSPS4 Promotes Sucrose Accumulation in Apple Fruit

Since the expression of the *MdSPS4* (MD10G1002500) gene was relatively high at 120 DAB and 150 DAB, the *MdSPS4* gene was selected for late functional validation in this paper. A viral vector-based method (vector pCAMBIA1300-GFP for overexpression and vector TRV for suppression) was used to test whether *MdSPS4* regulates sucrose accumulation in apple fruit. P1300–MdSPS4 and TRV–MdSPS4 viral constructs were produced and an empty vector (EV) used as control (Figure 9A). Then, *MdSPS4* expression levels were determined. It seemed *MdSPS4* suppression and overexpression decreased and increased the expression levels of *MdSPS4*, respectively (Figure 9B). In addition, the evaluation results of sucrose, fructose, and glucose content in apple fruit illustrated that the sucrose content of P1300–MdSPS4 was significantly higher than the control, while the sucrose content of TRV–MdSPS4 appeared lower than the apple fruit control (Figure 9C). Moreover, *MdSPS4* suppression in apple fruit reduced the fructose content, while *MdSPS4* overexpression had no significant effect on fructose content (Figure 9D). Interestingly, there was no significant difference in glucose content with *MdSPS4* overexpression or suppression compared to the control in apple fruit (Figure 9E). The results suggested that *MdSPS4* had a positive effect on sucrose accumulation in apple fruit.

To date, some work has revealed that *SPS* genes play a vital role in sucrose metabolism and plant growth. In potatoes, *SPS* overexpression can improve the yield of transgenic potatoes [48]. Moreover, *SPS* overexpression also alters the growth and development of transgenic tobacco [16], and enhances biomass production in *B. distachyon* [18]. The study also showed that the overexpression of the *SoSPS1* gene enhanced SPS activity and sucrose content in transgenic sugarcane leaves and stalks [18]. Meanwhile, *SoSPS1* overexpression increased soluble acid invertase (SAI) activity, which in turn led to the increase of glucose and fructose content in the leaves [18]. In Arabidopsis, *AtSPS4* mutation reduced the activity of sucrose phosphate synthase, but the effect is not significant, indicating that *AtSPS4* had little effect on sucrose accumulation in *Arabidopsis* [49]. Another study manifested that *SPS* overexpression contributed to increased sucrose unloading in tomato fruit [14]. In this study, overexpression and suppression of the *MdSPS4* gene were also able to significantly increase and decrease the sucrose content in apple fruit, respectively (Figure 9), which is consistent with the results of previous studies [18,49].

### 2.13. MdSPS4 Enhanced the Sucrose Content of Apple Callus in Response to Nitrogen

To elucidate the roles of *MdSPS4* in response to different nitrogen levels, The *MdSPS4* overexpression vectors of *pCAMBIA1300-MdSPS4-GFP* and pCAMBIA1300-GFP (EV) were transferred into ‘Orin’ callus and cultured in MS screening medium for 35 days. Fluorescence was observed in the *pCAMBIA1300-MdSPS4-GFP* callus under a stereomicroscope, indicating that the *MdSPS4* was successfully transferred into the ‘Orin’ callus (Figure 10F). Subsequently, the 15-day-old *MdSPS4-OE*, EV, and wild-type (WT) callus were grown in MS medium with different concentrations, in which potassium nitrate and ammonium nitrate components changed, while the other components remain unchanged (0-MS, 1-MS, and 1.5-MS). The results indicated that the *MdSPS4-OE* callus exhibited faster growth than the WT and EV in 1-MS and 1.5-MS (Figure 10A). In agreement with the phenotype, the *MdSPS4-OE* callus exhibited higher fresh weights than that of the WT and EV lines under 1-MS and 1.5-MS. Meanwhile, the *MdSPS4-OE* callus treated with 1-MS medium were better than that of 1.5-MS medium (Figure 10B). The results suggested that *MdSPS4* responded to nitrogen induction, and the growth of the ‘Orin’ callus was inhibited by either too high or too low nitrogen concentration. In addition, the sucrose, fructose, and glucose content in the ‘Orin’ callus was also determined after 0-MS, 1-MS, and 1.5-MS medium induction. The results showed that the sucrose content of the *MdSPS4-OE* callus lines was significantly higher than that of the EV and WT callus lines in 0-MS, 1-MS, and 1.5-MS medium, and the sucrose content of the 1.5-MS-induced apple callus was the highest (Figure 10C). Meanwhile, the fructose content of the *MdSPS4-OE* callus lines was significantly higher than WT and EV callus lines after 1.5-MS induction (Figure 10D), while the glucose content of the *MdSPS4-OE* callus lines was significantly higher than WT and EV callus lines after 1-MS induction (Figure 10E).

Yang et al. [20] showed the sucrose content of roots and shoots in winter wheat for the high nitrogen (7.5 mmol·L^−1^) was higher than those for the low nitrogen (1.25 mmol·L^−1^). Muchow et al. [50] indicated that the sucrose content varied in fresh sugarcane with different N supply and different development periods. Meanwhile, low N supply increased sucrose content in the early growth stage of sugarcane, while the sucrose content can decrease at the late growth stage under high nitrogen condition due to the decrease of stem dry matter content. In this study, *MdSPS4* overexpression in apple ‘Orin’ callus, not only notably increased sucrose content, but also responded to different nitrogen levels (Figure 10). With the higher nitrogen level, the sucrose content of apple callus was the higher, but too high nitrogen is not conducive to the growth of apple callus. This study hypothesizes that nitrogen promotes sucrose accumulation in apple callus by inducing *SPS* gene expression.

## 3. Materials and Methods

### 3.1. Plant Materials and Treatments

Eight-year-old ‘Oregon Spur Delicious’ (OSD)/MM.106 apple (*Malus domestica* Borkh.) grown in the apple garden at Maiji District, Tianshui City, Gansu Province (N 39°92′, E 116°40′, 1400 m a.s.l.) (Appendix A) were used as the test materials in 2019-2021. The apple garden region belongs to the continental semi-humid monsoon climate; the climate is mild and the average annual temperature is 10.7 ℃. The sunshine is sufficient and the average annual sunshine is 2090 h, annual precipitation is about 500 mm and the frost-free period is more than 170 days. 

Orchard soil nutrients were measured before nitrogen fertilizer was applied, taking the center of the ground corresponding to the canopy gap of fruit trees as the collection point. Soil samples of a 0–300 cm soil layer were collected by layers of soil drill, and soil samples were collected every 20 cm of soil layer. After the soil sample was pretreated, the contents of total nitrogen, fast-acting nitrogen, total phosphorus, fast-acting phosphorus, total potassium, and fast-acting potassium in soil were determined with the Kjeldahl method for nitrogen, the KCl extraction method, the ClO_4_-H_2_SO_4_-molybdenum-antimony resistance colorimetric method, the NaHCO_3_ extraction-molybdenum-antimony resistance colorimetric method, the NaOH fusion-flame photometry, and the NH_4_OAc extraction-flame photometry, respectively [51]. Orchard soil is loessal soil, and total nitrogen is 1.23 g/kg, fast-acting nitrogen is 118.75 mg/kg, total phosphorus is 1.36 g/kg, fast-acting phosphorus is 68.73 mg/kg, total potassium is 38.74 g/kg, and fast-acting potassium is 401.10 g/kg. 

Taking urea, commonly used in production as nitrogen fertilizer, the following four nitrogen levels were set: 0 kg·hm^−2^ (T0), 150 kg·hm^−2^ (T1), 300 kg·hm^−2^ (T2), and 600 kg·hm^−2^ (T3). There were three nitrogen application stages, including the germination stage (10 April 2019), the fruit expansion stage (25 June 2019), and one week before maturity (25 August 2019), accounting for 50%, 30%, and 20% of the annual nitrogen application amount, respectively. Three fruit trees with similar growth were selected as a replicate, and each treatment had nine trees. Fruit samples were collected at 30, 60, 90, 120, 135, and 150 days after bloom (DAB). 

‘Orin’ apple (*Malus domestica* Borkh.) callus was subcultured thrice at 3-week intervals on MS medium with 0.4 mg/L 6-BA and 1.5 mg/L 2,4-D at 25 °C in the weak light conditions.

### 3.2. Determination of Soluble Sugars and Organic Acids

Soluble sugars and organic acids were measured using an HPLC system (model 248, Waters, USA) equipped with an X BrigeTM amide column (3.5 mm, 4.6250 mm, USA) and a refractive index detector (model 2414, Waters) [52]. Determination of sorbitol, glucose, fructose, and sucrose content: 0.5 g apple fruit pulp was ground in liquid nitrogen and transferred to 10 mL centrifuge tube, 5 mL 80% ethanol was added, ultrasonically extracted at 35 °C for 20 min, centrifuged at 12,000 r/min for 15 min. The supernatant was transferred for rotary evaporation, the product was re-dissolved with 50% acetonitrile and finally filtered through a 0.22 µm organic membrane. Determination of malic acid, quinic acid, ascorbic acid, and citric acid content: 1.5 g of apple fruit pulp ground in liquid nitrogen and transfer to 10 mL centrifuge tube, 7.5 mL ultrapure water was added, centrifuged at 4 °C and 10,000 r/min for 10 min. Then the supernatant was transferred to a new centrifuge tube and filtered through a 0.45 μm aqueous membrane.

### 3.3. RNA Sequencing Analysis

A total of 18 independent RNA-seq libraries were constructed for the three biological replicates of the following two sampling time points and three nitrogen levels: 120 DAB (0, 300, and 600 kg·hm^−2^) and 150 DAB (0, 300, and 600 kg·hm^−2^). The prepared libraries were sequenced using the Illumina HiSeq 2000 system by Novogene (Beijing, China) [53]. All clean reads were mapped to the apple reference genome (https://iris.angers.inra.fr/gddh13) [54]. 

### 3.4. Differentially Expression Analysis

Differentially expressed genes (DEGs) were identified by FPKM (fragments per kilobase per million reads) values [55]. A statistical comparison of FPKM values between nitrogen fertilizer and non-nitrogen fertilizer fruit samples was performed using the web tool IDEG6 [56]. Differentially expression analysis of fruit samples was performed using the edgeR [57]. The *p*-value < 0.01 & Fold Change ≥1.5 was set as the threshold for differential expression significance.

### 3.5. Gene Functional Annotation

Gene function was annotated based on the following databases: Nr (NCBI non-redundant protein sequences); Nt (NCBI non-redundant nucleotide sequences); Pfam (Protein family); KOG/COG (Clusters of Orthologous Groups of proteins); Swiss-Prot (A manually annotated and reviewed protein sequence database); KO (KEGG Ortholog database) and GO (Gene Ontology).

### 3.6. Validation of DEGs by qRT-PCR 

The cDNA was synthesized using the manufacturer’s instructions for the Prime Script TM RT Master Mix (Perfect Real Time) kit (TaKaRa, Beijing, China). The expression patterns of 20 identified DEGs were verified by qRT-PCR, which was performed using the Takara SYBR Premix Ex Taq™ II kit (Takara) and run on the Light Cycler 96 instrument (Roche, Basel, Switzerland). The apple *GAPDH* gene was used as an internal reference to normalize all data. The relative expression levels of the DEGs were calculated using a 2^−∆∆CT^ method [58]. Primer sequences (Appendix A) were designed and synthesized by Sangon Biotech Co., Ltd. (Shanghai, China).

### 3.7. Construction of the Plasmid and Genetic Transformation

cDNA (full length) and antisense 5′-UTR sequences (partial) of *MdSPS4* (from ‘*Royal Gala*’ fruit) were used to construct *MdSPS4* overexpression and antisense repression vectors. These products obtained by RT-PCR assays were ligated into PCAMBIA1300 and TRV vectors with the *35S* promoter. The genetic transformation was performed using the method described by Xie et al. [59]. Primers are listed in Appendix A. 

### 3.8. Statistical Analysis 

All measurements were performed with three biological replicates. The software SPSS 22.0 (Stat Soft Inc., Palo Alto, CA, USA) was performed and *p* < 0.05 was the basis for statistical differences. Statistical analysis was performed with ANOVA and Duncan’s multiple comparison tests. 

## 4. Conclusions

This study has comprehensively characterized the physiological and transcriptomic analysis of apple in response to different nitrogen levels. RNA-seq analysis has revealed that sugar and acid metabolism-related pathways were significantly enriched, and pathways related genes (e.g., *HXK1, SPS4, SS2, PPC16-2,* and *MDH2* genes) exhibited significantly up- or down-regulated expression under different nitrogen levels. Meanwhile, the transcription factors AP2/ERF and NAC were strongly induced. Then, the function of *MdSPS4* was verified by transforming apple fruit and callus, the results indicate that *MdSPS4* could positively promote sucrose accumulation under nitrogen treatments. In conclusion, the combinational analysis of transcriptome and the functional validation of the *MdSPS4* gene provides new insights into apple responses at different nitrogen levels.

## Figures and Tables

**Figure 1 ijms-23-16073-f001:**
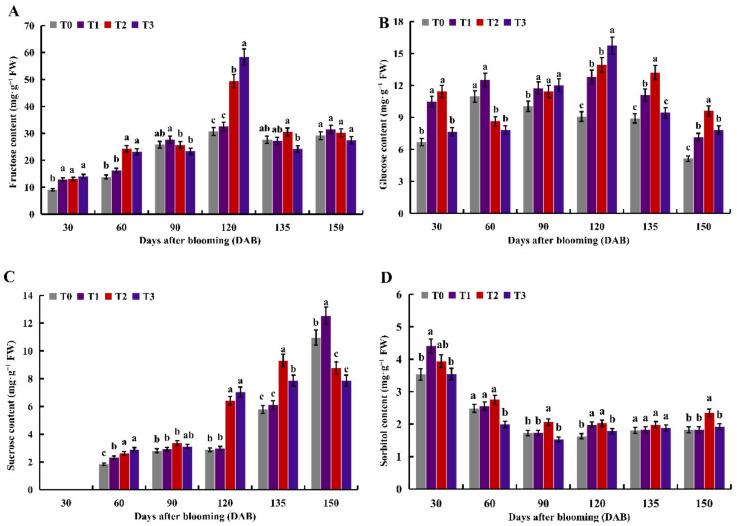
Effect of different nitrogen levels on soluble sugars in apple fruit. (**A**–**D**) Fructose, glucose, sucrose, and sorbitol content at 30, 60, 90, 120, 135, and 150 DAB, respectively. Different letters in the same period indicate significant differences between different treatments at the 0.05 level (*p* < 0.05).

**Figure 2 ijms-23-16073-f002:**
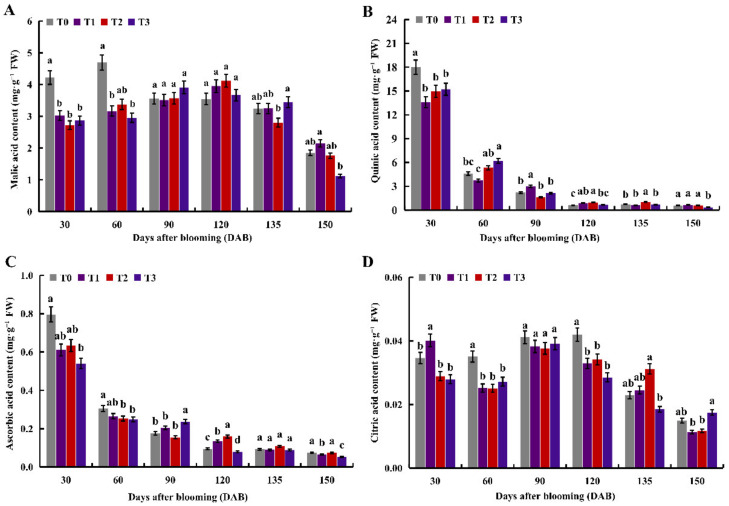
Effect of different nitrogen levels on organic acid in apple fruit. (**A**–**D**) Malic acid, quinic acid, ascorbic acid, and citric acid content at 30, 60, 90, 120, 135, and 150 DAB, respectively. Different letters in the same period indicate significant differences between different treatments at the 0.05 level (*p* < 0.05).

**Figure 3 ijms-23-16073-f003:**
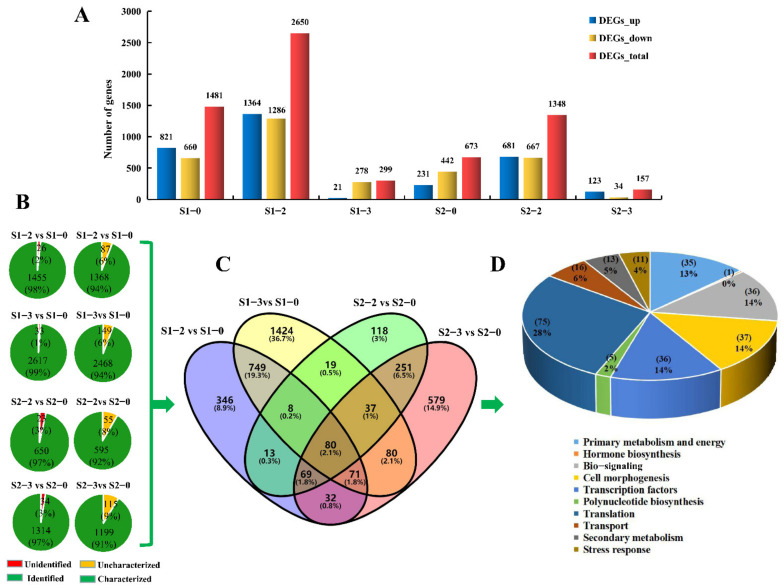
Distribution and classification of DEGs in apple fruit at different nitrogen levels (S1 and S2 periods). (**A**) The number of differential genes in S1–2 vs. S1–0, S1–3 vs. S1–0, S2–2 vs. S2–0, and S2–3 vs. S2–0. (**B**) Number and proportion of identified and characterized DEGs. (**C**) Venn diagram of characterized DEGs. (**D**) Functional classification of DEGs co-existed in the S1 and S2 periods.

**Figure 4 ijms-23-16073-f004:**
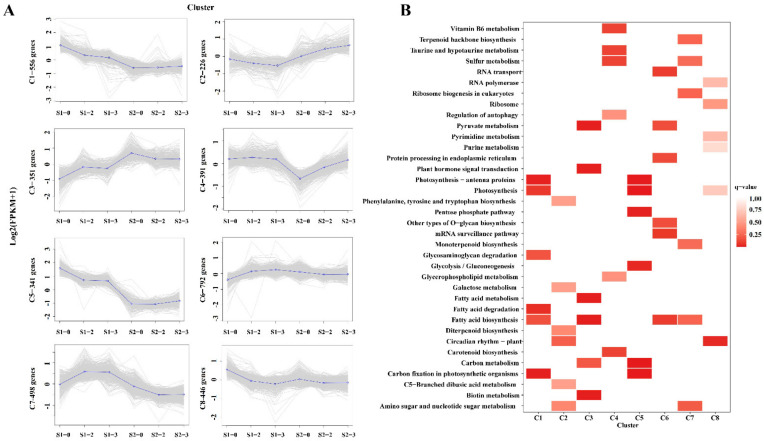
Expression pattern and functional analysis of the DEGs at different nitrogen levels. (**A**) Eight major clusters were identified. The X–axis represents different nitrogen treatments in the S1 and S2 periods. The y–axis represents the value of the relative expression level (log2 (FPKM + 1). (**B**) Functional analysis of the DEGs in S1 and S2 periods.

**Figure 5 ijms-23-16073-f005:**
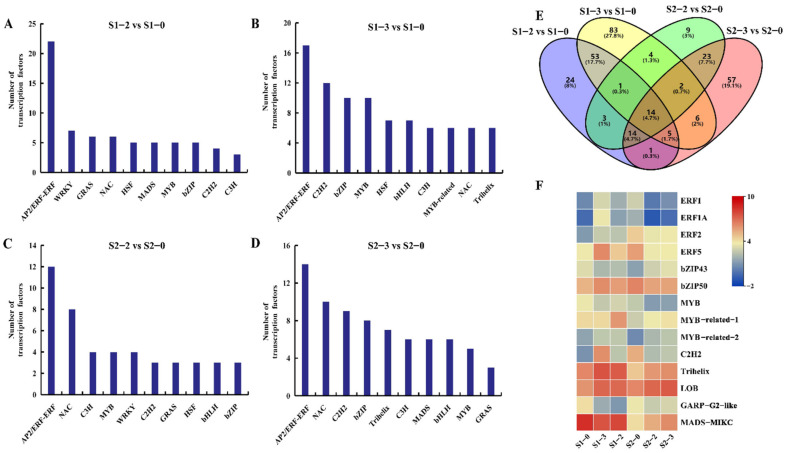
The distribution of differentially expressed transcription factors in apple fruit under different nitrogen levels. (**A**–**D**) The number of differentially expressed transcription factors in S1–2 vs. S1–0, S1–3 vs. S1–0, S2–2 vs. S2–0, and S2–3 vs. S2–0, respectively. (**E**) Venn diagram of differentially expressed transcription factors. (**F**) Heat map of differentially expressed transcription factors co-expressed in two periods.

**Figure 6 ijms-23-16073-f006:**
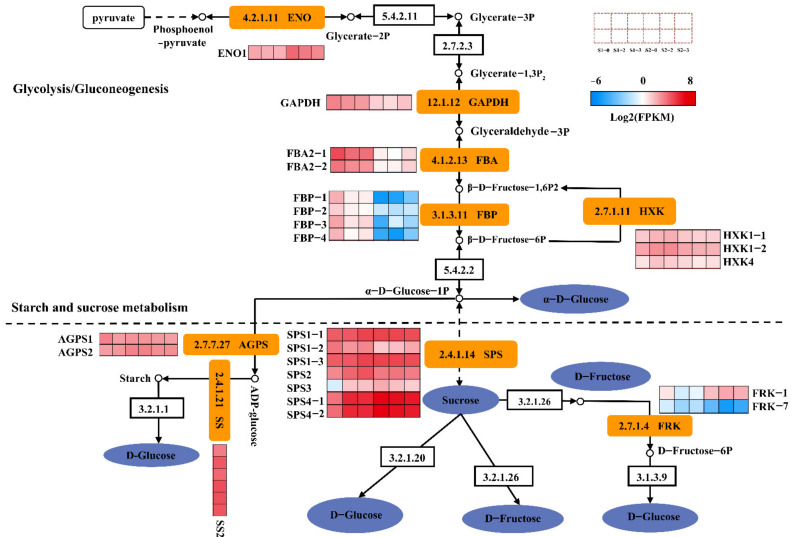
Regulation of sugar metabolism pathways in different nitrogen levels. The expression of DEGs is represented by Log2(FPKM). The color from blue to red represents the expression value from low to high.

**Figure 7 ijms-23-16073-f007:**
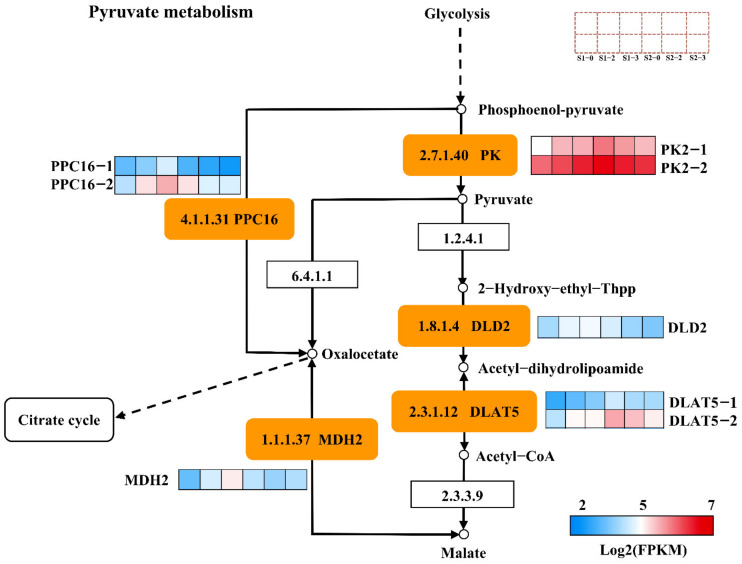
Regulation of pyruvate metabolism pathways in different nitrogen levels. The expression of DEGs is represented by Log2(FPKM). The color from blue to red represents the expression value from low to high.

**Figure 8 ijms-23-16073-f008:**
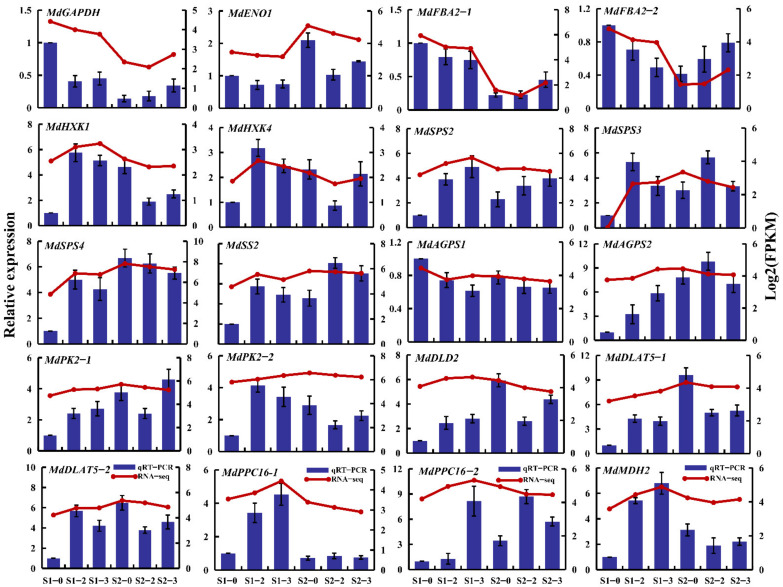
The qRT–PCR analysis of selected DEG genes in apple fruit under different nitrogen levels. Error bars represent standard errors of the relative expression levels mean values by qRT–PCR (*n* = 3) (left y–axis). Broken lines represent transcript levels (log2 FPKM) according to RNA–Seq (right y–axis).

**Figure 9 ijms-23-16073-f009:**
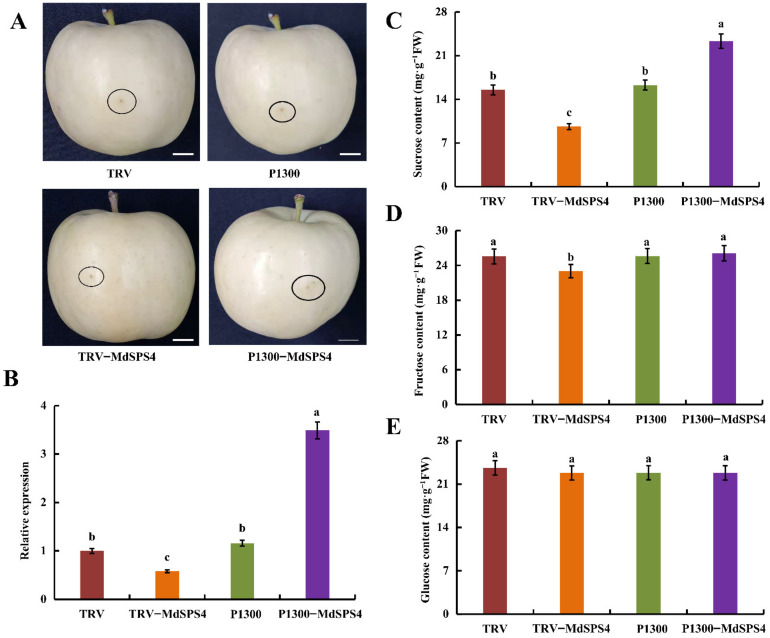
MdSPS4 promotes sucrose accumulation in apple fruit. (**A**) MdSPS4 transient expression in vectors. MdSPS4 cDNA fragment was ligated into the pCAMBIA1300–GFP vector. MdSPS4 antisense cDNA fragment was ligated into the TRV vector. Controls were empty vectors. Bars = 1 cm. (**B**) Relative expression of MdSPS4 at injection sites. (**C**) sucrose content in injected apple fruit. (**D**) fructose content in injected apple fruit. (**E**) glucose content in injected apple fruit. Error bars represent the means ± SD (*n* = 3) taken from three independent biological replicates. Different letters represent significant differences (LSD test, *p* < 0.05).

**Figure 10 ijms-23-16073-f010:**
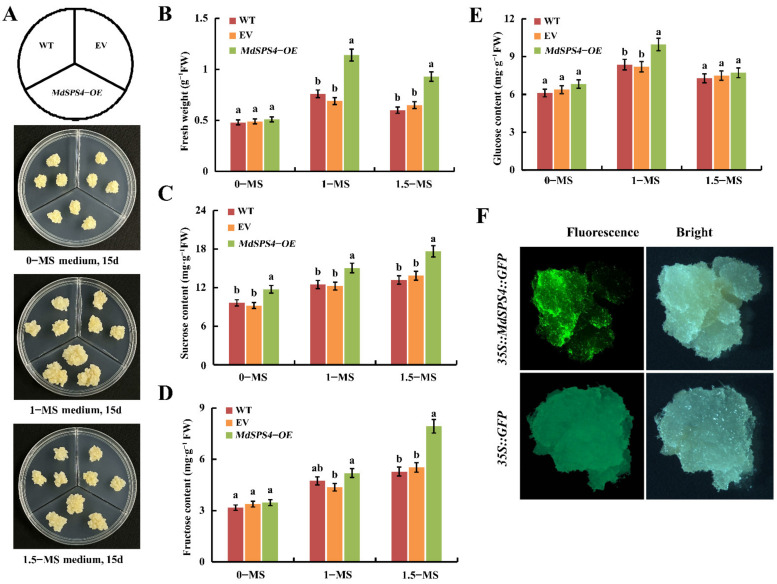
Effect of MdSPS4 on different nitrogen levels in transgenic apple calli. (**A**) The phenotypes of WT, EV, and transgenic apple calli were cultured on 0–MS, 1–MS, and 1.5–MS medium for 15 days. (**B**) Fresh weight. (**C**) Sucrose content. (**D**) Fructose content. (**E**) Glucose content. (**F**) Fluorescent photo of transgenic MdSPS4 apple calli. Error bars represent the means ± SD (*n* = 3) taken from three independent biological replicates. Different letters represent significant differences (LSD test, *p* < 0.05).

**Table 1 ijms-23-16073-t001:** The alignment statistics result with the reference gene for all samples.

DAB (Days)	N Fertilizer (kg·hm^−2^)	Clean Reads	Clean Bases	GC Content	%≥Q30	Mapped Reads	Unique Match
**120**	0	21,305,563	6,381,991,851	47.79%	93.32%	91.23	88.34
	300	20,527,705	6,147,942,123	47.80%	93.53%	91.62	88.39
	600	21,972,666	6,581,184,203	47.67%	93.37%	91.51	88.17
**150**	0	21,737,783	6,509,277,813	47.87%	93.58%	89.16	85.98
	300	21,471,510	6,428,687,591	47.99%	93.23%	88.36	85.04
	600	20,316,939	6,085,634,954	48.04%	93.37%	90.92	87.74

## Data Availability

Data will be made available on request.

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
