# Peer review of "New Insights into *MdSPS4*-Mediated Sucrose Accumulation under Different Nitrogen Levels Revealed by Physiological and Transcriptomic Analysis"

_ijms, 2022, doi:10.3390/ijms232416073_

Round 1

Reviewer 1 Report

The reviewed manuscript contains interesting research results. Please justify the use of the dose of 600 kg N ha-1 in the experiment. This is a very large dose of nitrogen and dangerous for the environment and consumers. Less harmful, but also very large is 300 kg N ha-1. What was the nitrate content in the soil and in the fruit? The authors omitted the aspect of food security.

Comments

Line 493, please specify the characteristics of the variety

Line 495, please post a map with the location of the research site

Line 496, there is no description of the course of weather conditions during the research period and in the multiyear

Line 497, please give a more detailed description of soil conditions and methods of analysis. What was the soil type? When were soil samples taken for testing?

Line 565, please provide full details of the statistical software manufacturer

References, please remove older publications older than 10 years, especially from the last century (1, 2, 15, 17, 50)

Author Response

Dear reviewers,

Thank you for your letter and for the comments concerning our manuscript entitled “New insights into MdSPS4-mediated sucrose accumulation under different nitrogen levels revealed by physiological and transcriptomic analysis’’ (Manuscript ID: IJMS-2061872). The comments have been valuable and have helped us improve our paper. Your feedback also served as an important guide to our investigation. We have studied the comments carefully and have applied corrections, which we hope would meet your approval.

Reviewer 1:

The reviewed manuscript contains interesting research results. Please justify the use of the dose of 600 kg N ha-1 in the experiment. This is a very large dose of nitrogen and dangerous for the environment and consumers. Less harmful, but also very large is 300 kg N ha-1. What was the nitrate content in the soil and in the fruit? The authors omitted the aspect of food security.

Answer: Thank you very much for your question. I will answer this question.

The loess plateau region has low organic matter content, poor soils, and is prone to nutrient loss. Under these ecological conditions, higher N fertilizer application can significantly increase the yield and quality of apple. Through the investigation of the data in previous years, it was found that after the application of 600 kg·hm-2 nitrogen fertilizer, the nitrate content in the apple fruit was less than 200 mg·kg-1, which was lower than the national standard of pollution-free fruit safety requirements.

Comments

Line 493, please specify the characteristics of the variety

Answer: Thank you very much for your comments, we have added it in introduction, see line 84-86.

Line 495, please post a map with the location of the research site

Answer: Thanks to your valuable comments, we have posted a map in Fig. S3

Fig. S3  The map with apple garden at Maiji District, Tianshui City, Gansu Province

Line 496, there is no description of the course of weather conditions during the research period and in the multiyear.

Answer: Thanks to your valuable suggestions, we have added it in line 495-499.

        Apple garden region belongs to the continental semi-humid monsoon climate, the climate is mild and the average annual temperature is 10.7 ℃, the sunshine is sufficient and the average annual sunshine is 2090 hours, annual precipitation is about 500 mm and the frost-free period is more than 170 days.

Line 497, please give a more detailed description of soil conditions and methods of analysis. What was the soil type? When were soil samples taken for testing?

Answer: Thanks to your valuable comments, we have added it in line 500-511.

Orchard soil nutrients were measured before nitrogen fertilizer was applied, taking the center of the ground corresponding to the canopy gap of fruit trees as the collection point, soil samples of 0-300 cm soil layer were collected by layers of soil drill, and soil samples were collected every 20 cm soil layer. After the soil sample is pretreated, the contents of total nitrogen, fast-acting nitrogen, total phosphorus, fast-acting phosphorus, total potassium, and fast-acting potassium in soil were determined by Kjeldahl method for nitrogen, KCl extraction method, ClO4-H2SO4-molybdenum-antimony resistance colorimetric method, NaHCO3 extraction-molybdenum-antimony resistance colorimetric method, NaOH fusion-flame photometry, and NH4OAc extraction-flame photometry, respectively [51]. Orchard soil is loessal soil, and total nitrogen is 1.23 g/kg, fast-acting nitrogen is 118.75 mg/kg, total phosphorus is 1.36 g/kg, fast-acting phosphorus is 68.73 mg/kg, total potassium is 38.74 g/kg, and fast-acting potassium is 401.10 g/kg.

Line 565, please provide full details of the statistical software manufacturer.

Answer: Thanks to your valuable suggestions, we have added it in line 558-559.

References, please remove older publications older than 10 years, especially from the last century (1, 2, 15, 17, 50).

Answer: Thanks to your valuable comments, we have remove older publications older than 10 years.

Reviewer 2 Report

Review report 

In general, This study is very strong and provided highly interesting data . It scientifically sounds with a great topic and a great impact on the field. The manuscript will be suitable for publication after a minor revision.

Detailed comments:

1-The English language and /writing style is fine needs some minor check spelling and grammar check

2-Please avoid using the personal pronouns (I, We, our) such as in line  567: we comprehensively characterized.

Abstract

_This section is missing the direct aim of the study. Please state the aim of the study clearly in this section.

Keywords:

-The keywords has been chosen very carefully and accurately . 

Introduction

-The introduction is very short and doesn’t provide enough background. It needs to be elongated and improved.

Materials and Methods

-This section is ok and the methods are adequate.

 Results:

The results are very interesting and well presented but some data needs a better discussion. 

Discussion:

_This section is fairly written but it can be improved for better explanation to the results.

I found that Figures 4,6,7, and 10 are not clearly discussed.

*The author is advised to combine the Results and Discussion in one section for better explanation and understanding to the  provided data especially because these data are very important and not easy for interpretation. 

** Also combining the results and discussion in one section beside the better organization it will also shorten the text and avoid unnecessary repetition. 

Conclusion :

This section is well written and the conclusion is supported by the results of this study and includes the most insights results.

References

This section is adequate, well written. And it is up To date . 

Author Response

Dear reviewers,

Thank you for your letter and for the comments concerning our manuscript entitled “New insights into MdSPS4-mediated sucrose accumulation under different nitrogen levels revealed by physiological and transcriptomic analysis’’ (Manuscript ID: IJMS-2061872). The comments have been valuable and have helped us improve our paper. Your feedback also served as an important guide to our investigation. We have studied the comments carefully and have applied corrections, which we hope would meet your approval.

Reviewer 2:

In general, This study is very strong and provided highly interesting data . It scientifically sounds with a great topic and a great impact on the field. The manuscript will be suitable for publication after a minor revision.

Detailed comments:

  • The English language and /writing style is fine needs some minor check spelling and grammar check.

Answer: Thank you very much for your comments, we have proceeded spelling and grammar check.

  • Please avoid using the personal pronouns (I, We, our) such as in line  567: we comprehensively characterized.

Answer: Thanks to your valuable suggestions, we have corrected personal pronouns (I, We, our) in paper.

Abstract

_This section is missing the direct aim of the study. Please state the aim of the study clearly in this section.

Answer: Thanks to your valuable suggestions, we have added it in line 10-12.

This study excavated crucial genes that regulated sugar metabolism in response to nitrogen in apple through physiology and transcriptome analysis, so as to lay a theoretical foundation for improving fruit quality.

Keywords:

-The keywords has been chosen very carefully and accurately. 

Introduction

-The introduction is very short and doesn’t provide enough background. It needs to be elongated and improved.

Answer: Thanks to your valuable suggestions, we have added it in line 77-90.

Apple is rich in minerals and vitamins and are planted in temperate regions of the world. Fruit quality is an indispensable indicator in apple production. The carbohydrates in apple are vital factors affecting the fruit quality, taste, and the formation of other secondary metabolites. In the loess plateau region, the main cultivation pattern is vigorous rootstocks grafting short branch varieties. Meanwhile, The planting area of the ‘Oregon Spur Delicious’ apple in the Tianshui City of Gansu Province is more than 66,666.67 hectares, which is the largest marshal apple cultivation area in the world. 'Oregon Spur Delicious' apple is the fifth generation of marshal apple with short shoots, which has the characteristics of bright skin color, crisp taste, tender meat, and strong aroma. The loess plateau region has low organic matter content, poor soils, and is prone to nutrient loss. Under these ecological conditions, higher N fertilizer application can significantly increase the yield and quality of apple. Therefore, it is very significant to elucidate the mechanism of different nitrogen levels on apple sugar metabolism.

Materials and Methods

-This section is ok and the methods are adequate.

 Results:

The results are very interesting and well presented but some data needs a better discussion. 

Answer: Thanks to your valuable comments, we have proceeded a better discussion of the results.

Discussion:

_This section is fairly written but it can be improved for better explanation to the results.

I found that Figures 4,6,7, and 10 are not clearly discussed.

Answer: Thanks to your valuable suggestions, we have proceeded clear discussion again.

*The author is advised to combine the Results and Discussion in one section for better explanation and understanding to the  provided data especially because these data are very important and not easy for interpretation. 

** Also combining the results and discussion in one section beside the better organization it will also shorten the text and avoid unnecessary repetition. 

Answer: Thanks to your valuable comments, we have combined the results and discussion in one section.

Conclusion:

This section is well written and the conclusion is supported by the results of this study and includes the most insights results.

References

This section is adequate, well written. And it is up To date 
